Histamine-2 receptor antagonist famotidine modulates cardiac stem cell characteristics in hypertensive heart disease

Saheera Sherin
Potnuri Ajay G.
Nair Renuka renukanairr52@gmail.com
Division of Cellular and Molecular Cardiology, Sree Chitra Tirunal Institute for Medical Sciences and Technology , Thiruvananthapuram , Kerala , India
Rehman Jalees
Electronic publication date: 2017 Oct 9
Publication date: 2017
Volume: 5
Electronic Location ID: e3882
Received 2017 May 9; Accepted 2017 Sep 12
Copyright: ©2017 Saheera et al.
Copyright year: 2017
Copyright holder: Saheera et al.
License: This is an open access article distributed under the terms of the Creative Commons Attribution License, which permits unrestricted use, distribution, reproduction and adaptation in any medium and for any purpose provided that it is properly attributed. For attribution, the original author(s), title, publication source (PeerJ) and either DOI or URL of the article must be cited.
License URL: https://creativecommons.org/licenses/by/4.0/

Keywords: Famotidine, Cardiac stem cells, Cardiac hypertrophy, Hypertension, Spontaneously hypertensive rat

Funding: Department of Atomic Energy India Department of Science and Technology This work was supported by the Department of Atomic Energy India and the Department of Science and Technology for the research Fellowship to Ms. Sherin. The funders had no role in study design, data collection and analysis, decision to publish, or preparation of the manuscript.

==============================
Background

Cardiac stem cells (CSCs) play a vital role in cardiac homeostasis. A decrease in the efficiency of cardiac stem cells is speculated in various cardiac abnormalities. The maintenance of a healthy stem cell population is essential for the prevention of adverse cardiac remodeling leading to cardiac failure. Famotidine, a histamine-2 receptor antagonist, is currently used to treat ulcers of the stomach and intestines. In repurposing the use of the drug, reduction of cardiac hypertrophy and improvement in cardiac function of spontaneously hypertensive rats (SHR) was reported by our group. Given that stem cells are affected in cardiac pathologies, the effect of histamine-2 receptor antagonism on CSC characteristics was investigated.

Methods

To examine whether famotidine has a positive effect on CSCs, spontaneously hypertensive rats (SHR) treated with the drug were sacrificed; and CSCs isolated from atrial appendages was evaluated. Six-month-old male SHRs were treated with famotidine (30 mg/kg/day) for two months. The effect of famotidine treatment on migration, proliferation and survival of CSCs was compared with untreated SHRs and normotensive Wistar rats.

Results

Functional efficiency of CSCs from SHR was compromised relative to that in Wistar rat. Famotidine increased the migration and proliferation potential, along with retention of stemness of CSCs in treated SHRs. Cellular senescence and oxidative stress were also reduced. The expression of H2R was unaffected by the treatment.

Discussion

As anticipated, CSCs from SHRs were functionally impaired. Stem cell attributes of famotidine-treated SHRs was comparable to that of Wistar rats. Therefore, in addition to being cardioprotective, the histamine 2 receptor antagonist modulated cardiac stem cells characteristics. Restoration of stem cell efficiency by famotidine is possibly mediated by reduction of oxidative stress as the expression of H2R was unaffected by the treatment. Maintenance of healthy stem cell population is suggested as a possible mechanism underlying the cardioprotective effect of famotidine.

Introduction

The discovery of cardiac stem cells challenged the notion of heart being a post-mitotic organ (Beltrami et al., 2003). Cardiomyocyte turnover ranges from 0.5–1% annually (Bergmann et al., 2009), implicating the involvement of cardiac stem cells in the maintenance of cardiac homeostasis. Protecting resident cardiac stem cells as a prelude to prevention of cardiac failure has not received much attention. The adverse microenvironment and the cycling of stem cells for replenishment of lost myocytes in the pathological heart can lead to stem cell aging. In the event of myocardial injury, cardiac stem cells mediate tissue repair and regeneration. The positive effect of stem cell transplantation for myocardial regeneration highlights the role of stem cells in tissue repair (Smits et al., 2005). Impaired efficiency of human cardiac stem cells has been documented in pathological conditions (Cesselli et al., 2011). Studies on the influence of cardioprotective drugs on cardiac stem cells are limited. Comparison of the effects of an angiotensin receptor blocker losartan and a beta-blocker metoprolol on left ventricular (LV) remodeling, in rat surgical model showed that along with improvement in LV function, the number of c-kit+ cells as well as expression of Ki-67 was increased by metoprolol but not losartan (Serpi et al., 2009). Therefore, selective cardio protective drugs are envisaged to modulate stem cell characteristics, to maintain a healthy stem cell pool.

Histamine, secreted by mast cells in the heart, has been implicated in cardiac diseases and the development of heart failure (Francis & Tang, 2006). Histamine receptors are present widely in the heart (Felix et al., 1988). We have reported that the H2 receptor antagonist famotidine promotes reverse cardiac remodeling in spontaneously hypertensive rats (SHRs) (Potnuri et al., 2016). Though impairment of CSCs is implicated in cardiac pathologies, modulation of stem cell characteristics by famotidine has not been reported. Hence, this study was carried out based on the assumption that famotidine will restore the cardiac stem cell efficiency, that is compromised in SHRs.

Materials and Methods

Experimental design

Six-month old SHRs were used as the experimental model. Twelve male SHRs were randomly assigned into two groups of six rats each. One group of untreated SHRs served as hypertensive control and the other group received a daily oral dose of 30 mg kg−1 day−1 of famotidine for two months. The response to treatment was evaluated by comparison with stem cell characteristics of untreated SHRs and normotensive Wistar rats (WST). The animals were housed at 22 °C, maintained on a 12 h light-dark cycle, fed with regular Rat Chow (Scientific Animal Food & Engineering, Augy, France) and had free access to drinking water. All animal procedures were approved by the Sree Chitra Tirunal Institute for Medical Sciences and Technology, Institutional Animal Ethics Committee, according to the Committee for the Purpose of Control and Supervision of Experiments on Animals (CPCSEA) Guidelines (Approval Reference No. B 2422012 XXI). The stem cell research was approved by The Sree Tirunal Institute for Medical Sciences and Technology, Institutional Committee for stem cell research (Approval No. SCT/IC-SCRT/01/Mar-2012).

Isolation, characterisation and expansion of CSCs

Atrial explants were established in 2% gelatin coated dishes supplemented with IMDM and 10% FBS. c-kit+ CSCs were isolated immunomagnetically using Easy Sep Magnet and Easy Sep FITC positive selection kit (Stem cell Technologies, Vancouver, Canada). The CSCs were expanded in culture medium containing IMDM along with 10% FBS, 10 ng/ml basic Fibroblast growth factor, 10 µl/ml Insulin- selenium-transferrin along with the antibiotics penicillin and gentamycin. The cells in passage 3 were assessed for their purity based on expression of cell surface markers –c-kit, CD 45, CD34 and CD 31 using flow cytometry and immunocytochemistry. The cells from the third passage were used for further studies.

Colony forming unit assay

Self renewing ability is a characteristic of stem cells. CSCs were plated at a cell density of 500 cells per 60 mm culture plate. After 14 days, the cells were washed with PBS and stained with 3% crystal violet in methanol for 30 min at room temperature. The colonies larger than 2 mm were counted.

Proliferation capacity

Cells were plated at a density of 10,000 cells per plate (Zhang et al., 2013). The cell number was obtained every 48 h till the 10th day. Growth kinetics was obtained by plotting the cell count. Growth rate (GR) and Population doubling time (PDT) were calculated using standard mathematical formulae. GR =ln(Nt∕N0)∕T, where T is the incubation time, N0 is the cell number at the beginning of the incubation time and Nt is the cell number at the end of the incubation time. Population doubling time was calculated using the formula, PDT = ln(2)/GR.

Migration ability

The migration potential was assessed by trans-well migration assay using serum as chemo-attractant. A total of 1 × 104 cells were seeded onto the upper chamber of the trans-well (BD Falcon, pore size—8 µm) in serum free medium. IMDM containing 10% serum was added in the lower chamber, where serum acted as the chemoattractant. Following incubation for 18 h, the cells on the upper surface of the membrane were removed and those that migrated to the lower surface were fixed with 4% paraformaldehyde, stained with crystal violet and counted.

Intracellular reactive oxygen species levels

The intracellular reactive oxygen species (ROS) levels in CSCs were determined from H2DCFDA fluorescence intensity (LeBel, Ischiropoulos & Bondy, 1992). CSCs were incubated for 10 min with 10 µM of DCFH2DA in DMSO. The fluorescence intensity of DCF was measured using Microplate reader. Fluorescence values were monitored by excitation at 498 nm and emission at 530 nm.

Senescent CSCs

The proportion of senescent cells was assessed based on the cytochemical analysis of senescence associated β-galactosidase staining using commercially available kits (Abcam, Cambridge, UK). The cells that stained positive for β-galactosidase were counted with the help of a light microscope. A minimum of five random fields from each dish with a minimum of 500 cells were counted and percentage of senescent cells was calculated.

Western blot analysis for expression of Histamine 2 receptor in CSCs

Twenty-five micrograms of protein from each sample were electrophoresed on 10% polyacrylamide SDS gel along with the histamine-2 receptor peptide. After transferring proteins to nitrocellulose membranes, they were incubated with the primary antibodies followed by appropriate secondary antibodies. Protein bands were visualized by chemi-luminescence and images were quantified using ImageJ software (Rasband, 1997–2017). The expression of target protein was normalised to their respective beta actins.

Statistical analysis

Values are presented as mean ± SD. One-way ANOVA was carried out and if there was significant variation between samples, Student t-test was used for evaluation of differences between samples. Results were considered statistically significant for p values less than 0.05.

Results

Isolation, culture and characterization of CSCs

Following famotidine treatment for 60 days, animals were sacrificed, hearts were dissected under aseptic conditions and atrial tissue was cultured as explants. Small, phase bright cells migrated from the explants within 14 days. The cells were trypsinised and subjected to immunomagnetic isolation to sort out the c-kit+ and lineage negative CSCs. The purity of the cultured CSCs was confirmed at passage 3 by flow cytometry and immunocytochemistry. The analysis revealed that 92 ± 3% cells were positive for c-kit and negative for hematopoietic, endothelial and mast cell markers CD45, CD31 and CD 34 respectively (Fig. 1). The cultured CSCs in the third passage were used for further experiments. The cells so isolated were able to form colonies when seeded at the rate of 0.5 cell/well in 96 well-plate further confirming their stem cell characteristics.

Figure 1 Characterization of the cultured CSCs at passage 3.

(A–D) Representative FACS images and (E) the percentage distribution of specific markers.

Effect of famotidine on self renewing ability of CSCs

Colony formation is characteristic of stem cells. The number of colonies formed from CSCs of SHR was 68% lower than that from WST. Famotidine treatment improved the self renewing capacity of CSCs and the values were comparable with that of WST (Fig. 2).

Figure 2 Effect of famotidine on self-renewing ability of atrial CSCs.

(A–C) Representative images and (D) Graphical representation of colony forming units (CFU) shown as number of colonies/culture plate (60 mm diameter). Data presented as mean ± SD. Variation was analyzed by ANOVA followed by Student t-test. (∗∗p < 0.01 SHR Vs WST; †p < 0.05 SHR Vs Famotidine) ANOVA p < 0.01 (n = 6/group).

Effect of famotidine on proliferation capacity of CSCs

Proliferation capacity as assessed by cell number, growth rate and PDT was significantly lower in SHR, reiterating the compromised efficiency of CSCs in hypertensive heart disease. The proliferation capacity of CSCs from SHRs improved significantly following treatment with famotidine (Fig. 3).

Figure 3 Effect of famotidine on growth kinetics, growth rate and population doubling time of atrial CSCs.

(A) Growth kinetics of CSCs represented as cell number*104 (B) Growth rate calculated as LogN of the ratio of cell number at two fixed time points (C) Population doubling time (PDT) represented as number of days. Data presented as mean ± SD. Variation was analyzed by ANOVA followed by Student t-test. (∗∗p < 0.01 SHR Vs WST; ††p < 0.01 & †p < 0.05 SHR vs. Famotidine) ANOVA p < 0.01 (n = 6/group).

Effect of famotidine on migration ability of CSCs

The migration rate of CSCs was 38% lower in SHR compared to WST, when assessed by trans-well assay. The lower infiltration towards the chemo-attractant serum reiterates their compromised efficacy. However, following treatment with famotidine, the migration potential improved by 15% indicating the beneficial effect of the drug (Figs. 4B–4E).

Figure 4 Effect of famotidine on intracellular reactive oxygen species (ROS) and migration potential of atrial CSCs.

(A) ROS levels in CSCs represented as H2DCFDA fluorescence intensity (B) Graphical representation and (C–E) Representative images of the migrated cells. Extent of migration is represented as mean number of cells/field. Data presented as mean ± SD. Variation was analysed by ANOVA followed by Student t-test. (∗∗p < 0.01 & ∗p < 0.05 SHR Vs WST; †p < 0.05 SHR vs. Famotidine) ANOVA p < 0.01 (n = 6/group).

Effect of famotidine on intracellular ROS levels of CSCs

Extracellular ROS, the major contributor towards hypertophic remodelling in SHR, influences the functional efficiency of CSCs by interacting with their microenvironment/niche. The intracellular ROS levels of CSCs as assessed by H2DCFDA fluorescence intensity, was significantly high in SHR and regressed on treatment with famotidine indicating the antioxidant effect of the drug (Fig. 4A).

Effect of famotidine on senescence of CSCs

The efficiency of CSCs is remarkably compromised in the presence of senescent cells. The proportion of senescent cells as assessed by SA-β-galactosidase staining was high in SHR compared to age-matched WST, but regressed significantly on treatment with famotidine (Figs. 5A–5D).

Figure 5 Effect of Famotidine on the senescence and Histamine-2-receptor expression of atrial CSCs.

(A) Graphical representation and (B–D) Representative images of senescent CSCs. Proportion of senescent cells is expressed as percentage of the total population. (E and F) Representative blots and graphical representation of the expression levels of H2R in atrial CSCs.Data presented as mean ± SD. Variation was analysed by ANOVA followed by Student t-test. (∗∗p < 0.01 SHR Vs WST; †p < 0.05 SHR Vs Famotidine) ANOVA p < 0.01 (n = 6/group). WST, Wistar rat, SHR, Untreated Spontaneously Hypertensive rat, Treated, SHR treated with Famotidine (30 mg/kg/day for two months).

Expression of Histamine 2 receptor in CSCs

Western blot analysis of the protein samples revealed the presence of histamine-2- receptor in CSCs. Upon normalization with β-actin, the expression levels of H2R was similar between age matched WST and SHR (Figs. 5E, 5F) The treatment did not alter the expression of the receptor.

Discussion

Hypertensive heart disease progresses through cardiac hypertrophy paving the way to heart failure, making it a major shareholder of all the mortalities worldwide. Cardiac hypertrophy involves the enlargement of cardiomyocytes along with increased fibrosis and decreased capillary density. Though cardiac remodelling starts as an adaptive response, in course of time it becomes maladaptive and the heart fails to counterbalance the pressure overload. Many physiological, cellular and molecular mechanisms play an interconnecting role in inducing remodelling. Of the several mechanisms predicted, oxidative stress is considered to be the critical determinant; being both the cause and consequence of pathological changes in cardiac hypertrophy. Studies from our laboratory has shown that oxidative stress precedes the hypertrophic remodeling in SHR and is evident as early as one month of age (Purushothaman et al., 2011).

The cardiac stem cells were discovered in 2003 and ever since, studies have been extensively carried out on c-kit+ cells. The endogenous stem cells in the heart play a critical role in maintaining cardiac homeostasis and regeneration (Bergmann et al., 2009), highlighting the significance of this minor population of cells. Studies have implicated the prominent role of cardiac stem cells in mediating tissue repair (Nadal-Ginard, Ellison & Torella, 2014). Very little information is available on the impact of pathological conditions on resident CSCs and the influence of cardioprotective drugs on modulating stem cell characteristics. The adverse microenvironment prevailing in the pathological heart can affect the efficient functioning of resident cardiac stem cells. Therefore, a conducive microenvironment is expected to protect the heart by modulation of stem cell characteristics. A study carried out in our laboratory showed that the anti-ulcer drug famotidine, a histamine-2 receptor antagonist, regressed hypertrophy with evident morphological and molecular changes (Potnuri et al., 2016). Other studies also support the role of histamine-2 receptor antagonism in prevention of heart failure (Francis & Tang, 2006). Finding improvement in cardiac structure and function in SHR, we chose to examine whether famotidine modulated the CSCs.

Heart was procured during the sacrifice of animals treated with famotidine for two months. Atria was dissected out and c-kit+ CSCs were immunomagnetically isolated from atrial explant cultures. The CSCs were then expanded in culture. Assessment of the purity and stemness of the CSCs at passage 3 confirmed the cardiac origin of these cells (Fig. 1) as they were positive for c-kit and negative for haematopoietic (CD 45), endothelial (CD 31) and mast cell (CD 34) markers (Okayama & Kawakami, 2006; Dahlin & Hallgren, 2015). Since the yield of CSCs from the primary isolation was low, the cells were expanded in culture to obtain cells in sufficient numbers for evaluation of the variables. Studies have shown that long term culture, up to passage 40 maintains the c-kit nature/stemness of CSCs (Miyamoto et al., 2010). The cells in the third passage were assessed for stem cell characteristics.

The ability to form colonies is a characteristic feature of stemness. CSCs from SHR, formed lower number of colonies compared to their age matched WST. Nevertheless, colony formation ability improved significantly in response to the treatment (Fig. 2) indicating restoration of stemness. Maintaining stem cells in their quiescent state within the niche will help to preserve the growth reserve of the heart. CSCs from SHRs had compromised proliferation capacity as evident from the decreased growth kinetics and increased PDT. Increase in growth kinetics (Figs. 3A–3C) upon treatment signifies the revival of proliferation efficiency and improved efficacy of the CSCs. Decreased proliferation and colony formation implies that CSCs from SHRs in the stable phase of hypertrophy are affected by maladaptive remodeling of the heart. Similar observations have been found when Adipose derived stem cells from Twitcher mice were compared with normal mice. The Adipose derived stem cells from diseased mice exhibited less self-replicating and proliferative capacity (Zhang et al., 2013).

Homing in of stem cells to the site of injury is an essential component for tissue repair. The functional efficiency of stem cells is determined by its ability to effectively invade and repair the damaged tissue (Guo et al., 2014). The deficient migratory capacity of CSCs with hypertensive heart disease implies functional impairment of CSCs for repair and regeneration upon demand. The migration potential of CSCs in SHRs was restored by famotidine (Figs. 4B–4E), denoting functional recovery. The improved invasive capacity is expected to aid the CSCs to effectively home in to the site of injury and replace the damaged myocytes; that occur at a relatively higher frequency in hypertrophic remodeling.

Stem cells, like somatic cells undergo aging process and express senescent markers (Chambers et al., 2007; Akunuru & Geiger, 2016). The presence of senescent cells in the niche can release paracrine signals that adversely affect the neighboring healthy cells (Schellenberg et al., 2011). The proportion of senescent cells was significantly higher in SHR compared with the age matched WST (Figs. 5A–5D). Famotidine-mediated decrease in senescence helps in preserving the stem cell pool.

The positive response of CSCs to famotidine can be mediated either by modulation of the microenvironment or the blockade of H2R, since famotidine is a H2R antagonist. H2R receptor is known to be present in stem cells and progenitor cells (Yamada et al., 2013; Liu et al., 2017), but the presence of histamine-2-receptors in CSCs has not been reported. This is the first study to report the presence of H2R in CSCs. Though the CSCs expressed H2R, the levels of receptor was not significantly different between SHR, WST and the treated group (Figs. 5E, 5F) implicating the role of other mechanisms for the improvement in the functionality of CSCs. This is also supported by the fact that stem cells residing within the niches are resistant to drugs due to the specific miroenvironment and hypoxic stability (Vinogradov & Wei, 2012; Alonso, Jones & Ghiaur, 2017). This can account for the lack of change in expression of H2R in SHR treated with famoidine (Figs. 5E, 5F).

The proposed mechanism of action of famotidine is the decrease in oxidative stress of the microenvironment. Famotidine treatment decreased myocardial oxidative stress and improved cardiac function, along with decrease of left ventricular wall thickness (Potnuri et al., 2016). Oxidative stress is the key regulator of multiple pathways involved in hypertrophy, and is detected at an early stage in SHR which prevails throughout the pathological remodelling (Purushothaman et al., 2011). An adverse microenvironment can contribute to the aging of stem cells. Enhanced oxidative stress in SHRs can be the major determinant for the inefficient functioning of stem cells. Though no reports on the intracellular ROS of CSCs is available, oxidative stress has been reported to affect the endothelial progenitor cells (Case, Ingram & Haneline, 2008). The increased ROS levels in CSCs from SHRs (Fig. 4A) suggest that intracellular oxidative stress can affect the overall efficiency of stem cells. The reduction of ROS upon treatment further supports the notion that the improved efficacy of CSCs is due to decreased oxidative stress both at the tissue and cellular levels. Hence, modulation of stem cell attributes by famotidine is possibly mediated by reduction of oxidative stress.

Conclusion

The fate of an organ is determined by the reserve of functionally efficient stem cells. In hypertensive heart disease, maintenance of a healthy stem cell population is expected to prevent progressive cardiac remodeling. Famotidine-mediated restoration of stem cell attributes therefore gains significance. The improvement in stem cell efficiency is possibly mediated by reduction of oxidative stress since the H2R expression in CSCs of SHRs was comparable to that of Wistar rats and was unaltered by treatment with famotidine. The modulation of stem cell efficiency by H2 receptor antagonism lends scope for further investigations enabling therapeutic application for prevention of progressive cardiac remodeling. Retrospective and prospective studies on the response to famotidine treatment has to be carried out for further validation of our findings.

Supplemental Information

Supplemental Information 1 Raw data

Click here for additional data file.

Additional Information and Declarations

Competing Interests

Author Contributions

Animal Ethics

Data Availability

The authors declare there are no competing interests.

Sherin Saheera conceived and designed the experiments, performed the experiments, analyzed the data, wrote the paper, prepared figures and/or tables.

Ajay G. Potnuri conceived and designed the experiments, performed the experiments, reviewed drafts of the paper.

Renuka Nair conceived and designed the experiments, contributed reagents/materials/analysis tools, reviewed drafts of the paper.

The following information was supplied relating to ethical approvals (i.e., approving body and any reference numbers):

All animal procedures were approved by the Sree Chitra Tirunal Institute for Medical Sciences and Technology, Institutional Animal Ethics Committee, according to the Committee for the Purpose of Control and Supervision of Experiments on Animals (CPCSEA) Guidelines (Approval Reference No. B 2422012 XXI). The stem cell research was approved by the Sree Tirunal Institute for Medical Sciences and Technology, Institutional Committee for stem cell research (Approval No. SCT/IC-SCRT/01/Mar-2012).

The following information was supplied regarding data availability:

The raw data is uploaded as a Supplemental File.

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
