# Peer review of "Histamine-2 receptor antagonist famotidine modulates cardiac stem cell characteristics in hypertensive heart disease"

_PeerJ, doi:10.7717/peerj.3882_

## Round 0.1 · original submission · Major Revisions

Although both reviews are brief, both reviewers highlighted the importance of showing raw data/representative images/flow cytometry data, the need for more experimental details and an expanded discussion of the findings. Reviewer 1 also asked key questions about potential mechanisms that need to be addressed.

Please provide a point-by-point response to the reviewer comments when you submit the revised manuscript.

Reviewer 1 ·

Basic reporting

No comment

Experimental design

Results seem like rather preliminary.

1. Provide representative images for figures 1a, 2a and 2c.
2. There is no mechanism explaining salutary effects of famotidine on CSCs. It would help if authors can provide a molecular basis for the observed effects.
3. Do CSCs express histamine receptor?
4. Authors note that histamine receptor is present primarily on mast cells, known to express c-kit. Can the authors provide results showing that their CSC population is negative for mast cells population tryptase.
5. What happens to wild type CSC proliferation and growth kinetics after treatment with famotidine in vitro.

Validity of the findings

The results are inconclusive and preliminary. More rigorous analysis required.

·

Basic reporting

The manuscript is clear but unfortunately includes no raw data. All results are presented as graphs. The authors should provide representative images for all three groups for Figures 1A, 2A, 2B and 2C. Furthermore, flow plots should be presented to justify the claim that the cells were 92+/-3% ckit+ and negative for CD45 and CD31.
The introduction is brief but adequate. The discussion should include more examples of comparision between this work and what is known in the field.

Experimental design

The experimental design is sound but the details of the methods are somewhat brief. For example, were the explants plated on coated flasks, what medium was used, which antibodies were used for flow, what were the cells stained with in the colony forming assay, how was the cell number obtained for the proliferation assay?
Minor comment - what is 'Age associated variation in fig 1D?

Validity of the findings

It is hard to assess the data when no raw images have been presented. If all the results are confirmed by raw data then the conclusions are sound.

Additional comments

This is an interesting piece of work but is marred by the lack of raw data and the brief discussion.

---

## Round 0.2 · Minor Revisions

Both reviewers were satisfied with the new data and felt that you sufficiently addressed all key points. At this point, we just need you to address the comments of the reviewer who recommended better quality images in Fig. 5, some methodological details that will help readers and overall language editing.

Reviewer 1 ·

Basic reporting

The manuscript is clear and significantly improved in the revised version.

Experimental design

The overall hypothesis is novel and interesting and addition of new methodology and results have improved the manuscript.

Validity of the findings

The findings are novel and well supported by the data.

Additional comments

The authors have addressed all the concerns raised.

·

Basic reporting

In this revision the authors now include improved discussion of their results and representative images to support their data. However, the images in figure 5 to show b-Gal staining and the Western blot in Fig 5E are not very convincing.

Experimental design

The methods are now improved with sufficient detail in most cases, although the quantities of FGF and insulin-selenium-transferrin are not given in 2.2

Validity of the findings

The data appears robust, although better images should be provided for fig 5 if possible.

Additional comments

The manuscript is much improved but requires some editing for English.

---

## Round 0.3 · Minor Revisions

Your revision has addressed the comments of the reviewer regarding the methodological details but the images in Figure 5 seem identical to ones in the previous version. Specifically, there were concerns about the quality of the Western blot in 5 E as well as quality of the immunostaining images in 5 B, C, D raised by the reviewer 2. The quality of the immunostaining and Western blot remain poor in this revision. Higher quality images for immunostaining and a new Western blot would address these concerns.

---

## Round 0.4 · accepted · Accept

Your latest revision has improved the quality of the images in Figure 5.